

# Peritumoral immune infiltrates in primary tumours are not associated with the presence of axillary lymph node metastasis in breast cancer: a retrospective cohort study

Carlos López[1,2,*], Ramón Bosch-Príncep[1,*], Guifré Orero[1], Laia Fontoura Balagueró[1], Anna Korzynska[3], Marcial García-Rojo[4], Gloria Bueno[5], Maria del Milagro Fernández-Carrobles[5], Lukasz Roszkowiak[3], Cristina Callau Casanova[1], M. Teresa Salvadó-Usach[1,2], Joaquín Jaén Martínez[1], Albert Gibert-Ramos[1], Albert Roso-Llorach[6], Andrea Gras Navarro[1], Marta Berenguer-Poblet[2,7], Montse Llobera[8], Júlia Gil Garcia[9], Bárbara Tomás[1], Vanessa Gestí[1], Eeva Laine[7], Benoît Plancoulaine[10], Jordi Baucells[11] and Maryléne Lejeune[1,2]

[1] Department of Pathology, Hospital de Tortosa Verge de la Cinta, Tortosa, Spain
[2] Campus Terres de l'Ebre, Universitat Rovira Virgili Tarragona, Tortosa, Spain
[3] Laboratory of Processing and Analysis of Microscopic Images, Nałęcz Institute of Biocybernetics and Biomedical Engineering, Warsaw, Poland
[4] Department of Pathology, Hospital Universitario Puerta del Mar, Cádiz, Spain
[5] VISILAB, Universidad de Castilla-La Mancha, Ciudad Real, Spain
[6] Institut Universitari d'Investigació en Atenció Primària Jordi Gol, Barcelona, Spain
[7] Department of Knowledge Management, Hospital de Tortosa Verge de la Cinta, Tortosa, Spain
[8] Department of Oncology, Hospital de Tortosa Verge de la Cinta, Tortosa, Spain
[9] Department of Surgery, Hospital Universitari de Girona Dr Josep Trueta, Girona, Spain
[10] UNICAEN, INSERM, ANTICEPE, Université de Caen Basse Normandie, Caen, France
[11] Department of Informatics, Hospital de Tortosa Verge de la Cinta, Tortosa, Spain
* These authors contributed equally to this work.

Corresponding authors
Carlos López,
cLópezp.ebre.ics@gencat.cat
Maryléne Lejeune,
mlejeune.ebre.ics@gencat.cat

## ABSTRACT

**Background.** The axillary lymph nodes (ALNs) in breast cancer patients are the body regions to where tumoral cells most often first disseminate. The tumour immune response is important for breast cancer patient outcome, and some studies have evaluated its involvement in ALN metastasis development. Most studies have focused on the intratumoral immune response, but very few have evaluated the peritumoral immune response. The aim of the present article is to evaluate the immune infiltrates of the peritumoral area and their association with the presence of ALN metastases.

**Methods.** The concentration of 11 immune markers in the peritumoral areas was studied in 149 patients diagnosed with invasive breast carcinoma of no special type (half of whom had ALN metastasis at diagnosis) using tissue microarrays, immunohistochemistry and digital image analysis procedures. The differences in the concentration of the immune response of peritumoral areas between patients diagnosed with and without metastasis in their ALNs were evaluated. A multivariate logistic regression model was developed to identify the clinical-pathological variables and the peritumoral

immune markers independently associated with having or not having ALN metastases at diagnosis.

**Results**. No statistically significant differences were found in the concentrations of the 11 immune markers between patients diagnosed with or without ALN metastases. Patients with metastases in their ALNs had a higher histological grade, more lymphovascular and perineural invasion and larger-diameter tumours. The multivariate analysis, after validation by bootstrap simulation, revealed that only tumour diameter (OR = 1.04; 95% CI [1.00–1.07]; $p = 0.026$), lymphovascular invasion (OR = 25.42; 95% CI [9.57–67.55]; $p < 0.001$) and histological grades 2 (OR = 3.84; 95% CI [1.11–13.28]; $p = 0.033$) and 3 (OR = 5.18; 95% CI [1.40–19.17]; $p = 0.014$) were associated with the presence of ALN metastases at diagnosis. This study is one of the first to study the association of the peritumoral immune response with ALN metastasis. We did not find any association of peritumoral immune infiltrates with the presence of ALN metastasis. Nevertheless, this does not rule out the possibility that other peritumoral immune populations are associated with ALN metastasis. This matter needs to be examined in greater depth, broadening the types of peritumoral immune cells studied, and including new peritumoral areas, such as the germinal centres of the peritumoral tertiary lymphoid structures found in extensively infiltrated neoplastic lesions.

## INTRODUCTION

Breast cancer (BC) patients with axillary lymph node (ALN) metastasis have a higher risk of distant metastases and death within 10 years of diagnosis. The primary cause of death in cancer patients is distant metastasis, most of which are incurable (*Siegel, Miller & Jemal, 2017*).

Immune cells are an important class of cells that are involved in tumoral progression (*Gardner & Ruffell, 2016*; *Hanahan & Coussens, 2012*; *Weber & Kuo, 2012*). The immune system protects against tumours, but cancer cells induce changes in the immune response, enabling them to evade immune destruction (*Corthay, 2014*). In most cases, the immune reaction against the tumour alone is ineffective at eliminating cancer cells due to the immunoediting and/or immunosubversion produced by the tumour. This is considered one of the emerging hallmarks of cancer (*Hanahan & Weinberg, 2011*). It is becoming clearer that distinct infiltrating cell types differ in their prognostic and predictive significance (*Fridman et al., 2011*). In BC, the intratumoral immune response has an important role in tumour progression, patient relapse and survival, among other processes (*De la Cruz-Merino et al., 2013*). In particular, tumour-infiltrating lymphocytes (TILs) in BC are of predictive and prognostic value, especially in triple-negative (TN) and human epidermal growth factor receptor (HER) 2-positive BC subtypes (*Loi et al., 2019*; *Salgado et al., 2015*). In fact, presence of TILs in the primary tumour significantly impacts the outcome of BC patients, especially when they have ALN metastasis at diagnosis (*Loi et al., 2019*). Nevertheless, only a few studies have evaluated the impact of either general TILs, by

haematoxylin and eosin (H&E) using Salgado's criteria (*Salgado et al., 2015*), or specific TIL subtypes in the peritumoral area (or invasive margin), by IHC. Two of them found no significant correlation between peritumoral immune infiltrates and clinical factors (*Acs et al., 2017*; *Al-Saleh et al., 2017*).

ALN status at the time of diagnosis is the most important prognostic indicator for women with BC (*Bernet Vegue, Cano Munoz & Pinero Madrona, 2012*). Moreover, ALN is the place to where the BC most often first disseminates (*Valente et al., 2014*). There is evidence of immune cell activation in invaded ALNs (*Gibert-Ramos et al., 2019*), and our group studied non-invaded ALNs of BC patients and identified several immune populations associated with the presence or absence of ALN metastasis at diagnosis (*López et al., 2020*), also highlighting the importance of the immune response of ALNs to patients' clinical outcome. Some studies have evaluated the possible association of the intratumoral immune response with ALN metastasis, but to our knowledge only one study has shown the peritumoral lymphocytic infiltrate to be an important predictive factor of the metastatic invasion of the ALN (*Bordea et al., 2012*). It is therefore of utmost importance to know whether the immune response in the peritumoral area of the primary tumour is associated with ALN metastasis in any way. The lack of research on this subject and the issues outlined above prompted us to study the association between the peritumoral immune response and the presence of ALN metastases at diagnosis.

## MATERIAL AND METHODS

### Tissue preparation and immunohistochemistry

This is a retrospective cohort study of 149 patients diagnosed with invasive BC of no special type in the Hospital de Tortosa Verge de la Cinta (HTVC), Spain, 75 of whom had ALN metastasis at diagnosis. The Ethics Committee of the Hospital Joan XXIII de Tarragona, Spain, approved the study (reference 22p/2011) and we followed the Strengthening the Reporting of Observational Studies in Epidemiology (STROBE) guidelines. Written informed consent was signed by all patients involved in the study, in accordance with Spanish law.

Two representative 2-mm tissue cylinders from the border of the tumour area of the biopsy were selected by a pathologist from the Pathology Department of the HTVC for the purpose of constructing tissue microarrays (TMAs). Ductal carcinoma in situ elements and tertiary lymphoid structures around the border of the tumour were excluded when selecting the areas from which cylinders were taken. Each TMA block contained 50 cylinders, giving 6 TMAs ((149 patients X 2 cylinders)/50 cylinders). Eleven slides were sectioned from each TMA in order to stain the 11 immune markers chosen for study (Fig. 1). TMA technology is of great value for analysing large numbers of cases, but it is clear that the degree of correlation between TMAs and whole-tissue sections is not ideal at the diagnostic level. Nevertheless, the use of TMAs with a large number of samples is widely considered to be adequate for research level (*Pinder et al., 2013*). In fact, as we mentioned in our previous report, in which we also used TMAs (*López et al., 2020*), a search using the terms "tissue microarray breast cancer immune" in PubMed identified more than 100 articles,

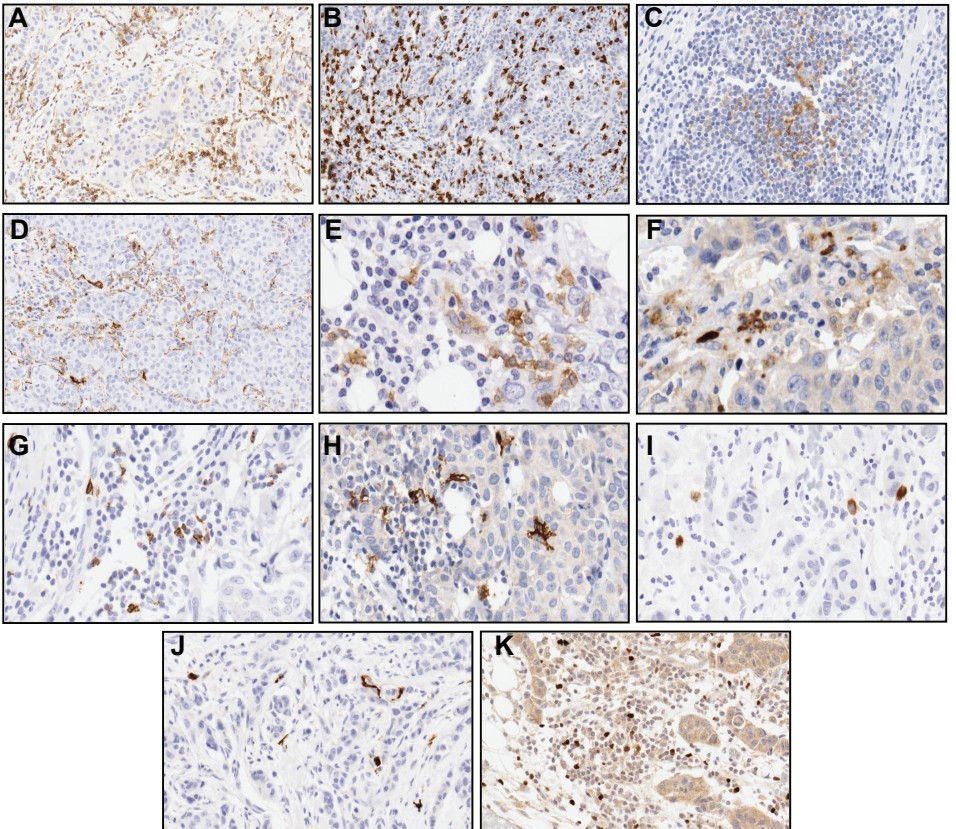

**Figure 1** **Immunohistochemical staining patterns of immune markers in formalin-fixed, paraffin-embedded sections.** Representative examples of membrane (A) CD4, (B) CD8, (C) CD21, cytoplasmic (D) CD68, (E) CD123, (F) LAMP3, membrane and/or cytoplasmic (G) CD57, (H) CD1a, (I) CD83, nuclear and/or cytoplasmic (J) S100 and nuclear (K) FOXP3 reactivity of the biomarkers (magnification 20X).

highlighting the widespread use of TMAs in studies into the evaluation of the immune system (*Pelekanou et al., 2018*; *Solinas et al., 2017*). Moreover, Salgado explained in his guide for evaluating TILs that results in TMAs have proven concordance with those of other studies (*Ali et al., 2014*; *Chavan, Ravindra & Prasad, 2017*; *Liu et al., 2014*; *Schalper et al., 2014*), which makes them a good option for rapid evaluations (*Salgado et al., 2015*).

The selection of immune markers for the present work was based on the findings of other studies that have demonstrated the various immune cell populations evaluated (lymphocytes, macrophages and the DC) to be associated with BC patient outcome (*De Melo Gagliato et al., 2017*; *Stovgaard et al., 2018*; *Zhao et al., 2017*). Immune cells were immunohistochemically detected on each slide using the following primary antibodies: T helper lymphocytes (anti-CD4, clone 4B12, Dako, Santa Clara, CA, USA), cytotoxic T lymphocytes (anti-CD8, clone C8/144B, Dako, Santa Clara, CA, USA), natural killers (anti-CD57, clone NK1, Zymed, Thermo Fisher Scientific, Waltham, MA, USA), regulatory T cells (anti-FOXP3, clone 236A/E7, CNIO, Madrid, Spain), macrophages (anti-CD68,

clone KP1, Dako, Santa Clara, CA, USA), follicular DC (anti-CD21, clone 1F8, Dako, Santa Clara, CA, USA), Langerhans DC (anti-CD1a, clone 010, Dako, Santa Clara, CA, USA), plasmacytoid DC (anti-CD123, clone 6H6, eBioscience, San Diego, CA, USA), interdigitant DC (anti-S100, polyclonal, Leica Microsystems GmbH, Wetzlar, Germany), LAMP3 DC+ (anti-CD208, polyclonal, Proteintech, Rosemont, IL, USA), mature DC (anti-CD83, clone1H4b, Leica Microsystems GmbH, Wetzlar, Germany). The ENDVISION$^{TM}$ FLEX method (Dako, Santa Clara, CA, USA) was applied to stain the slides, using the chromogen diaminobenzidine (DAB) as a substrate. The samples were counterstained with haematoxylin.

## Slide digitization and image evaluation

We used whole-slide imaging (WSI), a method that is replacing the microscope for classical diagnosis in some centres (*Pantanowitz et al., 2013*), to analyse digital images. We obtained the images in TIFF format by scanning the 66 stained slides with an Aperio ScanScope XT scanner at 40X magnification at a resolution of 0.25 µm/pixel. We extracted each cylinder of the original WSI from each TMA as a single image using an automatic tool developed by members of our team (*Roszkowiak & López, 2016*). The tissue cylinder areas and the stained areas of immune markers in each image were evaluated using our own digital image analysis procedures (*López et al., 2020*), which enable the number and density (in µm$^2$) of pixels of the positive-stained areas for each immune marker, and the area of each cylinder included in the TMAs, to be calculated (*Callau et al., 2015*). The concentration of each immune marker was calculated as the percentage of positive-stained areas of each immune marker relative to the whole area of the cylinder, as previously described (*López et al., 2020*).

## Clinical and pathological variables

To determine which clinical and pathological variables, in addition to the peritumoral immune response, could also be associated with the presence of metastasis in the ALN at diagnosis, the following data were collected from the patients' clinical records: age, tumour diameter, lymphovascular invasion (LVI), perineural invasion (PNI), histological grade, oestrogen receptor status (ER), progesterone receptor status (PR), HER–2 status, proliferation index (Ki67), menopausal status and molecular profile.

## Statistical analysis

Differences in immune response marker concentrations between patients diagnosed with and without metastasis in their ALN were evaluated using the Mann–Whitney U test. The quantitative clinical and pathological variables in the two groups of patients were compared using the Mann–Whitney U test or Student's unmatched samples $t$-test (age and tumour diameter). In order to identify disproportionate frequencies of combinations of categories of the clinical and pathological variables we performed chi-squared test or Fisher's exact tests.

A univariate logistic regression analysis was carried out for each variable to evaluate its association with the presence or absence of ALNs with metastasis. Two multivariate logistic regression models were then developed to identify which of the clinical and pathological

variables, and which of the immune cell populations present in the peritumoral area, were associated with the presence of ALN metastases at diagnosis. The Hosmer–Lemeshow test was used to estimate the goodness of fit of all the variables considered in the multivariate analyses. The area under the curve (AUC) and the receiver-operating characteristic (ROC) curve were also derived to estimate the sensitivity and specificity of each model. We can consider that the present study, which featured 75 events and yielded final multivariate models comprising three independent variables, had an adequate sample size for a reliable multivariate analysis as previously suggested (*Peduzzi et al., 1996*). In the first model, all variables with a significance of $p \leq 0.1$ in the univariate logistic regression analyses were considered when deriving the multivariate model 1. In the second model, all the variables with a significance of $p \leq 0.3$ were included. Each model was validated using two statistical techniques: (1) bootstrap simulation, carried out with IBM SPSS Statistics 21.0 (IBM, Armonk, NY, USA), based on 10,000 random samples; and (2) the multiple imputation method available in the IBM SPSS statistical application. This method replaces missing values of a specific variable by using linear regression to calculate values from others in the dataset. Failing to deal with missing data is a problem because it leads to a reduction in the statistical power of the model and can produce biased estimates.

## RESULTS

Table 1 shows the differences in the clinical and pathological variables between patients with and without metastasis in their ALNs at diagnosis. Patients with metastases in their ALNs have a higher histological grade, more LVI and PNI, and larger-diameter tumours.

Table 2 shows the differences in the percentages of the immune populations in the peritumoral regions between patients diagnosed with and without metastatic ALN. There were no differences in the median concentration in the immune populations between the two groups of patients evaluated by the WSI and the digital image procedures.

We next identified the variables associated with the presence of metastasis in the ALN at diagnosis from the univariate logistic regressions and using several multivariate logistic regression models. In the first model, we only included those variables that were significant or had a value of $p \leq 0.1$ in the univariate analysis. In this case, only the histological grade and the presence of LVI were independently associated with the presence of ALN metastases at diagnosis (Table 3). None of the immune variables was included in multivariate model 1, as none of the peritumoral immune populations showed any association with ALN metastases at diagnosis or a value of $p \leq 0.1$ in the univariate model. The Hosmer–Lemeshow test of this first model indicated an excellent goodness of fit to the final model ($p = 0.798$). Nagelkerke's R-squared was 0.577, indicating that around 60% of the variance of the dependent variable (presence of metastasis in the ALN) was explained by the model. The logistic regression model had a sensitivity of 78.4%, a specificity of 86.5% and an AUC of 0.898 (Fig. 2A, black line). The bootstrap validation of multivariate logistic regression model 1 identified tumour diameter (OR=1.04; 95% CI [1.00–1.07]; $p = 0.026$), LVI (OR = 25.42; 95% CI [9.57–67.55]; $p < 0.001$) and histological grades 2 (OR=3.84; 95% CI [1.11–13.28]; $p = 0.033$) and 3 (OR = 5.18; 95% CI [1.40–19.17]; $p = 0.014$) as

**Table 1 Differences in the clinical and pathological variables between patients with and without ALN[+] at diagnosis.**

| | Patients without ALN[+] at diagnosis ($n = 74$) | Patients with ALN[+] at diagnosis ($n = 75$) | p |
|---|---|---|---|
| **Age (years)** | 61.3 (10.7) | 59.7 (12.0) | 0.394[a] |
| **Tumour diameter (mm)** | 15.0 (12.3) | 22.0 (13.0) | <0.001[b] |
| **LVI** | | | |
| Yes | 9 (12.2%) | 57 (77.0%) | <0.001[c] |
| No | 65 (87.8%) | 17 (23.0%) | |
| **PNI** | | | |
| Yes | 10 (13.5%) | 30 (40.5%) | <0.001[c] |
| No | 64 (86.5%) | 44 (59.5%) | |
| **Histological grade** | | | |
| 1 | 27 (36.5%) | 8 (10.7%) | <0.001[c] |
| 2 | 30 (40.5%) | 34 (45.3%) | |
| 3 | 17 (23.0%) | 33 (44.0%) | |
| **ER expression** | | | |
| Positive | 55 (74.3%) | 53 (70.7%) | 0.617[c] |
| Negative | 19 (25.7%) | 22 (29.3%) | |
| **PR expression** | | | |
| Positive | 50 (67.6%) | 41 (54.7%) | 0.106[c] |
| Negative | 24 (32.4%) | 34 (45.3%) | |
| **HER-2 amplification** | | | |
| Amplified | 18 (24.7%) | 20 (26.7%) | 0.780[c] |
| Non-amplified | 55 (75.3%) | 55 (73.3%) | |
| **Ki 67 degree** | | | |
| Low | 26 (35.1%) | 16 (21.9%) | 0.120[c] |
| Medium | 26 (35.1%) | 25 (34.3%) | |
| High | 22 (29.8%) | 32 (43.8%) | |
| **Menopausal status** | | | |
| Pre-menopausal | 8 (11.6%) | 15 (22.1%) | 0.101[c] |
| Post-menopausal | 61 (88.4%) | 53 (77.9%) | |
| **Molecular profile** | | | |
| Luminal A | 47 (64.3%) | 46 (61.4%) | 0.967[c] |
| Luminal B | 10 (13.7%) | 10 (13.3%) | |
| HER-2 | 8 (11.0%) | 10 (13.3%) | |
| Triple-negative | 8 (11.0%) | 9 (12.0%) | |

**Notes.**

ALN[+], metastatic axillary lymph node; LVI, lymphovascular invasion; PNI, perineural invasion; ER, oestrogen receptor; PR, progesterone receptor; HER, human epidermal growth factor receptor.

[a]The data and statistical tests summarised in the table are the mean (standard deviation) for the $t$-Student test.

[b]The median (interquartile range) for the Mann-Whitney U test

[c]The number of patients (percentage) in each category for the chi-squared or Fisher's exact test

being significant factors. The AUC of validated model 1 was 0.898, with a sensitivity of 81.1% and a specificity of 86.5% (Fig. 2B, black line). All variables included in model 1, illustrated in Table 3, were retained in the validated model, the latter also showing tumour diameter to be associated with metastasis in the ALN at diagnosis (Table 4).

**Table 2  Differences in the percentages of the immune populations in the peritumoral regions between patients diagnosed with and without ALN$^+$.**

|  | Patients without ALN$^+$ at diagnosis ($n = 74$) | Patients with ALN$^+$ at diagnosis ($n = 75$) | p |
|---|---|---|---|
| **Peritumoral** |  |  |  |
| CD4 | 1.6 (3.4) | 1.7 (3.6) | 0.782 |
| CD8 | 1.4 (2.8) | 2.0 (3.3) | 0.386 |
| CD57 | 0.2 (0.5) | 0.3 (0.8) | 0.099 |
| FOXP3 | 0.1 (0.2) | 0.1 (0.2) | 0.598 |
| CD21 | 0.000 (0.005) | 0.000 (0.001) | 0.405 |
| CD68 | 2.4 (2.3) | 2.7 (3.0) | 0.221 |
| CD1a | 0.1 (0.3) | 0.1 (0.2) | 0.133 |
| CD123 | 0.00 (0.09) | 0.00 (0.08) | 0.377 |
| S100 | 0.3 (0.5) | 0.3 (0.4) | 0.516 |
| LAMP3 | 0.005 (0.021) | 0.000 (0.034) | 0.127 |
| CD83 | 0.1 (0.1) | 0.1 (0.2) | 0.139 |

Notes.

ALN$^+$, metastatic axillary lymph node.
The values in the table are the median (interquartile range) of the percentage of positive area expressed for each marker. The differences between groups were evaluated using the Mann-Whitney U test.

We derived a second multivariate model that included those variables with a level of significance of $p \leq 0.3$ (Table 3). The immune populations considered when generating this second model were CD8+ T lymphocytes, CD68+ macrophages, CD1a+ Langerhans DC, S100+ interdigitant DC and CD123+ LAMP3 DC. In the end, the multivariate model comprised only the LVI (with a wider CI than in model 1) and the CD68+ macrophages. Nagelkerke's R-squared was 0.789, and the Hosmer–Lemeshow test of this second model indicated excellent goodness of fit to the final model ($p = 0.794$) once again. Compared with model 1, this second model was more sensitive (88.4%), less specific (63.3%) and had a lower AUC (0.842) (Fig. 2A, red line). Nevertheless, the bootstrap validation of model 2 only retained LVI (OR=24.93; 95% CI [9.54–65.19]) as a significant factor; while the CD68+ macrophage factor was dropped. The AUC of this validated model was 0.852 (Fig. 2B, red line), with a sensitivity of 78.3% and specificity of 86.7%.

As an alternative validation system to the first and second models we used multiple imputation for the immune populations. The validation of the first model gave the same results with respect to the OR (Table 4), AUC, sensitivity and specificity as in the unvalidated first model (Fig. 2C, black line). After imputing the missing data, none of the immune populations was statistically significant or yielded a value of $p \leq 0.1$, so the same variables as in model 1 were included for the purpose of validation. In the validation of the second model using a threshold of $p \leq 0.3$, LVI, histological grade and the tumour diameter were included in the model, but none of immune populations was retained. The validated model had a sensitivity of 79.7%, a specificity of 87.8% and an AUC of 0.898 (Fig. 2C, red line).

To summarize, in three of the four validations of the two models, the histological grade and LVI were factors independently associated with ALN metastasis, and in two of the validations tumour diameter was also included. None of the four validated models featured

**Table 3** Univariate and multivariate analyses of variables associated with ALN$^+$ at diagnosis.

| | Univariate OR (95% CI) | p | Multivariate model 1 OR (95% CI) | p | Multivariate model 2 OR (95% CI) | p |
|---|---|---|---|---|---|---|
| Age (years) | 0.99 (0.96–1.02) | 0.392 | | | | |
| Tumour diameter (mm) | 1.06 (1.03–1.09) | 0.001 | | | | |
| **LVI** | | | | | | |
| Present | 24.2 (10.0-58.5) | <0.001 | 25.3 (9.54-66.90) | <0.001 | 372.28 (13.22–10485.09) | 0.001 |
| Absent | – | – | – | – | – | – |
| **PNI** | | | | | | |
| Present | 4.36(1.94-9.83) | <0.001 | | | | |
| Absent | – | – | | | | |
| **Histological grade** | | | | | | |
| 3 | 6.55 (2.45-17.5) | <0.001 | 5.13 (1.40-18.94) | 0.014 | | |
| 2 | 3.82 (1.51-9.69) | 0.005 | 3.83 (1.11–13.16) | 0.033 | | |
| 1 | – | – | | | | |
| **ER** | | | | | | |
| Positive | 0.83 (0.41-1.71) | 0.617 | | | | |
| Negative | – | – | | | | |
| **PR** | | | | | | |
| Positive | 0.58 (0.30-1.13) | 0.108 | | | | |
| Negative | – | – | | | | |
| **HER-2** | | | | | | |
| Amplified | 1.11 (0.53-2.33) | 0.780 | | | | |
| Non-amplified | – | – | | | | |
| **PI (Ki 67)** | | | | | | |
| High | 2.36 (1.04-5.40) | 0.041 | | | | |
| Med | 1.56 (0.68-3.58) | 0.292 | | | | |
| Low | – | – | | | | |
| **Menopausal status** | | | | | | |
| Pre-menopausal | 0.46 (0.18-1.18) | 0.106 | | | | |
| Post-menopausal | – | – | | | | |
| **Molecular profile** | | | | | | |
| HER-2 | 1.28 (0.46-3.52) | 0.637 | | | | |
| TN | 1.15 (0.41-3.24) | 0.792 | | | | |
| Luminal B | 1.02 (0.39-2.69) | 0.965 | | | | |
| Luminal A | – | – | | | | |
| **Peritumoral immune markers** | | | | | | |
| CD4 | 1.03 (0.92–1.15) | 0.644 | | | | |
| CD8 | 1.07 (0.95–1.21) | 0.238 | | | | |
| CD57 | 1.04 (0.96–1.13) | 0.320 | | | | |
| FOXP3 | 0.76 (0.15–3.75) | 0.736 | | | | |
| CD21 | 0.67 (0.10–4.51) | 0.682 | | | | |
| CD68 | 1.09 (0.94–1.27) | 0.249 | | | 2.14 (1.20–3.83) | 0.010 |
| CD1a | 0.67 (0.32–1.38) | 0.272 | | | | |
**Table 3** (*continued*)

| | Univariate OR (95% CI) | p | Multivariate model 1 OR (95% CI) | p | Multivariate model 2 OR (95% CI) | p |
|---|---|---|---|---|---|---|
| CD123 | 1.98 (0.10–38.65) | 0.654 | | | | |
| S100 | 0.64 (0.28–1.44) | 0.282 | | | | |
| LAMP3 | 5.73 (0.25–129.3) | 0.272 | | | | |
| CD83 | 3.54 (0.54–23.28) | 0.188 | | | | |

**Notes.**

ALN+, metastatic axillary lymph node; OR, odds ratio; CI, confidence interval; LVI, lymphovascular invasion; PNI, perineural invasion; ER, oestrogen receptor; PR, progesterone receptor; HER, human epidermal growth factor receptor; PI, proliferation index.

Multivariate models 1 and 2 include variables from the univariate analyses with a level of significance of $p \leq 0.1$ and $p \leq 0.3$, respectively.

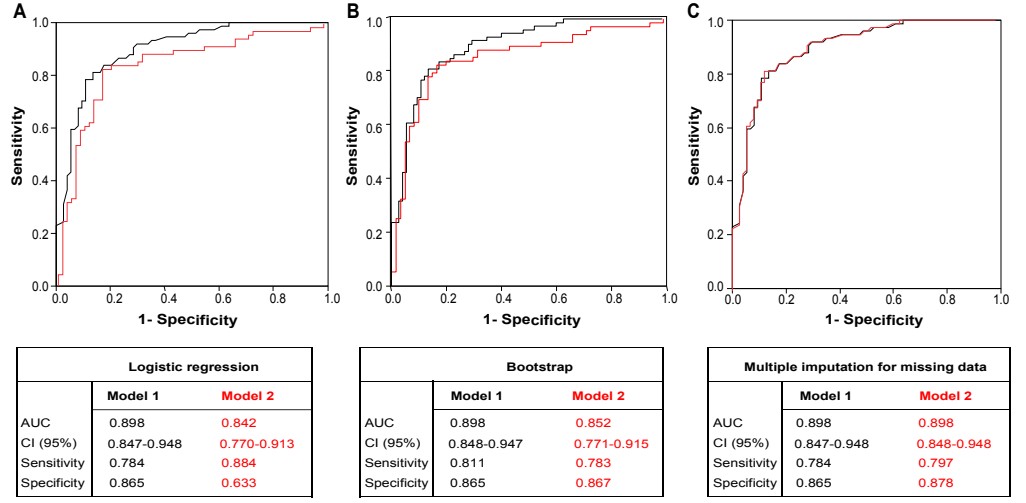

**Figure 2** **Receiver-operating characteristic (ROC) curves.** ROC curves of (A) the two multivariate logistic regression models; (B) the two validated multivariate logistic regression models derived by the bootstrap method; and (C) the two validated multivariate logistic regression models derived by multiple imputation. Values of the area under the curve (AUC) and the 95% confidence interval (CI) for each original model and validated model are presented.

any of the peritumoral immune populations associated with the presence of ALN metastasis at diagnosis. Only in the second model were CD68+ macrophages associated with ALN metastasis, although even this term was dropped from both validated models (using bootstrapping and multiple imputation methods). With the analysis of our data we mostly observe that the association of CD68 and ALN metastasis (outcome) is not statistically significant and there is more evidence in favour (4 validation + model including variables with $p \leq 0.1$) than against (model including variables with $p \leq 0.3$). Therefore, we assume that the evidence for the association of CD68 and metastasis is not robust enough to conclude that it is a significant variable in our models.

## DISCUSSION

In the present study, 11 immune markers in the peritumoral area were examined using WSI, TMA and digital image analysis procedures, to evaluate their association with the

**Table 4  Univariate and multivariate analyses (multiple imputation).**

| | Univariate OR (95% CI) | p | Multivariate model 1 OR (95% CI) | p | Multivariate model 2 OR (95% CI) | p |
|---|---|---|---|---|---|---|
| **Age (years)** | 0.99 (0.96–1.02) | 0.392 | | | | |
| **Tumour diameter (mm)** | 1.06 (1.03–1.09) | 0.001 | 1.04(1.00–1.07) | 0.057 | 1.04 (1.00–1.08) | 0.038 |
| **LVI** | | | | | | |
| Present | 24.2 (10.0-58.5) | <0.001 | 25.27(9.54–66.90) | <0.001 | 27.58 (9.55-79.67) | <0.001 |
| Absent | – | – | – | – | – | – |
| **PNI** | | | | | | |
| Present | 4.36 (1.94-9.83) | <0.001 | | | | |
| Absent | – | – | | | | |
| **Histological grade** | | | | | | |
| 3 | 6.55 (2.45-17.5) | <0.001 | 5.13(1.39-18.94) | 0.014 | 6.43 (1.49-27.75) | 0.013 |
| 2 | 3.82 (1.51-9.69) | 0.005 | 3.83(1.11-13.16) | 0.033 | 4.48 (1.14-17.64) | 0.032 |
| 1 | – | – | – | – | – | – |
| **ER** | | | | | | |
| Positive | 0.83 (0.41-1.71) | 0.617 | | | | |
| Negative | – | – | | | | |
| **PR** | | | | | | |
| Positive | 0.58 (0.30-1.13) | 0.108 | | | | |
| Negative | – | – | | | | |
| **HER-2** | | | | | | |
| Amplified | 1.11 (0.53-2.33) | 0.780 | | | | |
| Non-amplified | – | – | | | | |
| **PI (Ki 67)** | | | | | | |
| High | 2.36 (1.04-5.40) | 0.041 | | | | |
| Medium | 1.56 (0.68-3.58) | 0.292 | | | | |
| Low | – | – | | | | |
| **Menopausal status** | | | | | | |
| Pre-menopausal | 0.46 (0.18-1.18) | 0.106 | | | | |
| Post-menopausal | – | – | | | | |
| **Molecular profile** | | | | | | |
| HER-2 | 1.28 (0.46-3.52) | 0.637 | | | | |
| TN | 1.15 (0.41-3.24) | 0.792 | | | | |
| Luminal B | 1.02 (0.39-2.69) | 0.965 | | | | |
| Luminal A | – | – | | | | |
| **Peritumoral immune markers** | | | | | | |
| CD4 | 1.04 (0.93–1.16) | 0.497 | | | | |
| CD8 | 1.09 (0.97–1.22) | 0.148 | | | | |
| CD57 | 1.04 (0.97–1.12) | 0.250 | | | | |
| FOXP3 | 1.00 (0.998–1.001) | 0.555 | | | | |
| CD21 | 0.80 (0.13–5.00) | 0.815 | | | | |
| CD68 | 1.12 (0.97–1.29) | 0.132 | | | | |

**Table 4** (*continued*)

|  | Univariate OR (95% CI) | p | Multivariate model 1 OR (95% CI) | p | Multivariate model 2 OR (95% CI) | p |
|---|---|---|---|---|---|---|
| CD1a | 0.76 (0.50–1.14) | 0.757 | | | | |
| CD123 | 1.71 (0.24–12.25) | 0.594 | | | | |
| S100 | 0.82 (0.57–1.17) | 0.272 | | | | |
| LAMP3 | 7.08 (0.36–138.89) | 0.197 | | | | |
| CD83 | 4.46 (0.69–28.86) | 0.116 | | | | |

**Notes.**

ALN[+], metastatic axillary lymph node; OR, odds ratio; CI, confidence interval; LVI, lymphovascular invasion; PNI, perineural invasion; ER, oestrogen receptor; PR, progesterone receptor; HER, human epidermal growth factor receptor; PI, proliferation index.

Multivariate models 1 and 2 include variables from the univariate analyses with a level of significance of $p \leq 0.1$ and $p \leq 0.3$, respectively.

presence of metastasis in the ALN at diagnosis of BC patients. Even though BC is not a highly immunogenic tumour (*Gingras et al., 2015*), several studies have demonstrated the importance of the immune response in tumour progression and patient outcome (*De la Cruz-Merino et al., 2013*; *Denkert et al., 2010*; *Gu-Trantien & Willard-Gallo, 2013*; *Loi, 2013*; *Loi et al., 2013*). Indeed, in TN and HER2-positive BC, TILs are predictive of neoadjuvant therapy and prognostic in patients treated with chemotherapy (*Dieci et al., 2018*; *Loi et al., 2019*).

### Immune response and ALN metastasis

In BC, metastasis in ALNs plays a key role in spreading tumoral cells to other parts of the body (*Ran et al., 2010*); in fact, several types of intratumoral cells are known to be linked to the presence of metastasis in the ALN at diagnosis: cytotoxic and helper T lymphocytes (*La Rocca et al., 2008*; *Matkowski et al., 2009*), T regulatory lymphocytes (*Gokmen-Polar et al., 2013*; *Miyan et al., 2016*; *Nakamura et al., 2009*), macrophages (*Jubb et al., 2010*; *Mansfield et al., 2012*; *Shiota et al., 2016*) and dendritic cells (*Mansfield et al., 2011*; *Treilleux et al., 2004*). In addition, our group previously found the CD21+ follicular DC immune population in the intratumoral region to be associated with the presence of ALN metastasis. In the same study, we compared the immune populations of the non-metastatic ALN in patients diagnosed with or without ALN metastasis. We found that higher concentrations of CD68+ macrophages and of S100+ interdigitant DC in the non-metastatic ALNs were associated with the presence of ALN metastasis at diagnosis. On the other hand, higher concentrations of CD123+ plasmacytoid DC were found to be a factor protecting against ALN metastasis (*López et al., 2020*). Since the immune populations could be associated with ALN metastasis in locations other than the intratumoral region, we decided to evaluate the peritumoral area as well.

### Peritumoral immune infiltrates in BC

Little is known about the peritumoral immune infiltrates, and, especially, their links with ALN status. Bordea et al. showed BC to be more aggressive and associated with an increased rate of sentinel lymph node metastasis in patients with peritumoral TILs (*Bordea et al., 2012*). However, they did not use H&E or Salgado's criteria; the latter were published in 2015, and have since become established as the gold standard for TILs assessment (*Salgado et al., 2015*). Specific types of immune cells have barely been studied in the

peritumoral areas of BC patient samples. CD4+ follicular helper T cells are associated with peritumoral tertiary lymphoid structures (TLS) (*Gu-Trantien et al., 2013*). High levels of expression of intratumoral CD8+ TILs are significantly associated with the overall survival (OS) of luminal B patients treated with anthracycline-based neoadjuvant chemotherapy, but the peritumoral fraction is not (*Al-Saleh et al., 2017*). Conversely, Vgenopoulou et al. investigated peritumoral CD8 and CD57 markers, and found increased numbers of CD8+ cells in ALN+ patients, but no difference in the abundance of the CD57 marker (*Vgenopoulou et al., 2003*). Liu et al. reported high levels of peritumoral FOXP3+ to be a predictor for chemotherapy of HER2-positive patients. In addition, the latter article is the only one, to our knowledge, to report an association between the immune response in the peritumoral area of the primary tumour and ALN metastasis, showing positive correlation between peritumoral FOXP3+ and positive nodal status (*Liu et al., 2014*). A study analysing CD68+ macrophage cells showed that these cells were more likely to be present in the intratumoral area than in the peritumoral area, although its correlation with ALN status was not studied. The study was also limited by its low number of specimens (*Carpenco, 2019*). Controversially, Heiskala et al. found CD68+ infiltrations to be more abundant in the peritumoral than the intratumoral area, but CD68 frequency was not correlated with ALN status (*Heiskala et al., 2019*). Next, considering DCs, Bell et al. reported immature CD1a+ to be retained intratumorally, and mature CD83+ DCs to be confined to peritumoral areas in patients with BC; nonetheless, the study was performed with only 32 samples and no statistical analyses were carried out (*Bell et al., 1999*). However, our study revealed no association between the immune infiltrates of the peritumoral area and the presence of ALN metastases at diagnosis, which is in line with the findings of the few studies that have investigated the relation of the immune populations in the peritumoral areas with ALN status.

## Clinical and pathological factors associated to ALN metastasis

Most of the studies on the clinical, molecular and pathological/histological factors associated with ALN metastasis were carried out a long time ago (*Ahlgren et al., 1994*; *Chadha et al., 1994*; *Noguchi et al., 1993*). We found primary tumour size, histological grade and LVI to be histological factors significantly associated with the presence of ALN metastases at diagnosis in two of four validations. These findings are consistent with the well-established pathological characteristics associated with ALN metastasis reviewed by *Patani, Dwek & Douek (2007)* and described in subsequent publications (*Reynders et al., 2014*; *Yoshihara et al., 2013*). In the present study, model 1 had the best AUC when validated and so had the best predictive capability. The validated first model included the LVI, primary tumour size and histological grade as independent factors associated with having ALN metastasis, and showed very good sensitivity and specificity. Patani et al. highlighted several works showing the values of the sensitivity and specificity of the multivariate models using tumour size, histological grade and LVI (*Patani, Dwek & Douek, 2007*). In some of these studies, sensitivity was related to tumour size, or the histological grade was higher than in our case, but conversely they showed very low specificity (*Barth, Craig & Silverstein, 1997*; *Reynders et al., 2014*; *Silverstein, Skinner & Lomis, 2001*). On the other hand, several studies

employing models using LVI were very specific but not at all sensitive (*Chadha et al., 1994*; *Tan et al., 2005*). Our model gave a very good balance between sensitivity and specificity; in fact, the AUC of the validated first model was almost 0.9, which implies that our model can correctly classify around 90% of patients, and shows that these variables are closely associated with the presence of ALN metastasis at diagnosis.

Furthermore, *Tseng et al. (2014)* and *Grigoriadis et al. (2018)* have demonstrated that other histological features may be associated with the presence of ALN with metastasis at diagnosis (lymphocytic lobulitis, size and number or location of germinal centres, etc.). These variables could also be used to evaluate BC patient outcome. In fact, another study showed that the presence of CD4+ T cells, localized in the germinal centres of peritumoral TLS found in extensively infiltrated neoplastic lesions, predicted better disease outcome among BC patients (*Gu-Trantien & Willard-Gallo, 2013*). Indeed, TLS have recently proven to be relevant in patients' survival and immunotherapy response in other cancer types like sarcoma and melanoma (*Cabrita et al., 2020*; *Helmink et al., 2020*; *Petitprez et al., 2020*). Nevertheless, we did not study the TLS on this occasion, although in the future it would be worthwhile evaluating them in the context of the immune response.

## CONCLUSIONS

The present study aimed to determine whether the immune response in the peritumoral area of the primary BC tumour was associated with ALN metastasis at diagnosis. We studied 11 populations of immune infiltrates in the peritumoral areas by immunohistochemistry and did not find any association with the presence of metastases in the ALNs at diagnosis in BC patients. This does not rule out the possibility that other peritumoral immune populations are associated with ALN metastasis. This matter needs to be studied further, broadening the types of peritumoral immune cells studied and including new peritumoral areas, such as the germinal centres of the peritumoral TLS.

## ACKNOWLEDGEMENTS

The authors would like to thank María del Mar Barberá, Anna Curto, Noelia Burgués, Ainhoa Montserrat, Eduard Nolla, Maria Fortuny, Sandra Bages, Mireia Sueca and Marc Iniesta for their skillful technical assistance, and Anna Carot for her excellent secretarial work.

### Funding

This work was funded by projects PI11/0488 and PI13/02501 of the Institute of Health Carlos III, which is the main public research body that funds, manages and carries out biomedical research in Spain, and co-financed with European Union ERDF funds (European Regional Development Fund). It was also supported by the Project AIDPATH FP7-PEOPLE Project ID: 612471. The funders had no role in study design, data collection and analysis, decision to publish, or preparation of the manuscript.

## Grant Disclosures

The following grant information was disclosed by the authors:

Institute of Health Carlos III.

European Union ERDF funds (European Regional Development Fund).

Project AIDPATH FP7-PEOPLE Project: 612471.

## Competing Interests

The authors declare there are no competing interests.

## Author Contributions

- Carlos López conceived and designed the experiments, performed the experiments, analyzed the data, prepared figures and/or tables, authored or reviewed drafts of the paper, and approved the final draft.
- Ramón Bosch-Príncep, Marcial García-Rojo and Joaquín Jaén Martínez conceived and designed the experiments, performed the experiments, authored or reviewed drafts of the paper, and approved the final draft.
- Guifré Orero performed the experiments, analyzed the data, authored or reviewed drafts of the paper, and approved the final draft.
- Laia Fontoura Balagueró and M. Teresa Salvadó-Usach analyzed the data, prepared figures and/or tables, and approved the final draft.
- Anna Korzynska, Gloria Bueno, Maria del Milagro Fernández-Carrobles, Lukasz Roszkowiak, Albert Gibert-Ramos and Montse Llobera performed the experiments, authored or reviewed drafts of the paper, and approved the final draft.
- Cristina Callau Casanova performed the experiments, prepared figures and/or tables, and approved the final draft.
- Albert Roso-Llorach analyzed the data, authored or reviewed drafts of the paper, and approved the final draft.
- Andrea Gras Navarro and Marta Berenguer-Poblet analyzed the data, prepared figures and/or tables, authored or reviewed drafts of the paper, and approved the final draft.
- Júlia Gil Garcia performed the experiments, prepared figures and/or tables, authored or reviewed drafts of the paper, and approved the final draft.
- Bárbara Tomás, Vanessa Gestí, Benoít Plancoulaine and Jordi Baucells performed the experiments, authored or reviewed drafts of the paper, and approved the final draft.
- Eeva Laine performed the experiments, prepared figures and/or tables, and approved the final draft.
- Maryléne Lejeune conceived and designed the experiments, performed the experiments, analyzed the data, authored or reviewed drafts of the paper, and approved the final draft.

## Human Ethics

The following information was supplied relating to ethical approvals (i.e., approving body and any reference numbers):

The study was approved by the Ethics Committee of the Hospital Joan XXIII de Tarragona, Spain (Reference 22p/2011).

## Data Availability

Raw data is available as a Supplemental File.

## Supplemental Information

Supplemental information for this article can be found online at http://dx.doi.org/10.7717/peerj.9779#supplemental-information.

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
