# Peer review of "Peritumoral immune infiltrates in primary tumours are not associated with the presence of axillary lymph node metastasis in breast cancer: a retrospective cohort study"

_PeerJ, doi:10.7717/peerj.9779_

## Round 0.1 · original submission · Minor Revisions

Pay especially careful attention to the comments from Reviewer 2, who felt these changes were major rather than minor, and address them to the best of your ability.

Reviewer 1 ·

Basic reporting

see below

Experimental design

See below

Validity of the findings

See below

Additional comments

The paper entitled “Peritumoral immune infiltrates in primary tumors are not associated with axillary lymph node metastasis in breast cancer: a retrospective cohort study” by C Lopez et al. reports the retrospective analysis of 11 immune markers in a series of 150 ductal invasive breast equally distributed into N+ and N- cases (75 each). The markers (CD4, CD8, CD57, CD21, CD68, CD83, CD123, FOXP3, S100, LAMP3) were assessed by immunohistochemistry on tissue specimens taken at the periphery of the tumor, using tissue micro-arrays. The statistical analysis showed a significant link between the classical prognostic factors (tumor size, histological grade, vascular or peri-neural invasion), but no significant association was observed between lymph node status and expression level of any of the 11 markers analyzed.
This negative study has been conducted by authors familiar with the topic and, based on this, one may deduce that the methodology was correct. The paper is rather elusive (22 lines of results). Two weaknesses are to be underlined. (I) The significant association reported with the classical factors does not derive from parameters assessed in the present study; (II) no data of the immune parameter assessed within the tumor tissue is provided, that would have validated the methodology and the significance of the present series. The reader is invited to understand that this part of the study has already been reported in the paper recently published by the same group (Am J Pathol 2020). Tissue micro-arrays provide a limited evaluation of the tumor microenvironment, especially at the periphery of tumors, and any positive association would have supported the validity of the present study.
On the whole, the paper is acceptable a short note providing a negative result of limited value which significance might be deduced from the previous paper.

Reviewer 2 ·

Basic reporting

The authors post an interesting question of whether peritumoural infiltration in breast cancer influences the risk of developing distant metastasis. The study encompasses 150 primary breast carcinomas of NST and 11 immune markers were tested by IHC. No difference in peritumoural infiltrates was detected.

In the introduction, the authors need to mention Salgado's criteria and put their analyses in view of Salgado's TILs assesment.

Experimental design

More detail needs to be supplied on how the percentage of positivity for IHC immune markers was defined.
The result section needs rewriting and expansion - each marker needs to be discussed and multiple models could be tested to see if a combination of markers is risk-predictive.
ER-positive and ER_negative breast cancers have very different trajectories of developing breast cancer and thus different models need to be used for analyses.
Molecular subtypes would indicate different numbers of HER2-positive and ER-negative cases then the IHC - thus Table 1 needs to be checked if it correct?

Validity of the findings

Only if TILs are captured according to Salgado's criteria, these findings can be interpreted and their value assessed.

Additional comments

NA

·

Basic reporting

The manuscript is quite comprehensive, including the well decribed groups of patients and the statistics. Sharing the anonymized patients data is appreciated (filename "Patients data_PeerJ.sav"). Having opened this file, patient number 124 is missing, so I can see 149 patients in sum, which would differ from the number of 150 provided in the abstract and the materials and methods part (line 34). The lost patient should be added in the file or the number n should be corrected to 149 and numbers n should be provided in the tables of the manuscript as there might be <nan> (or empty) values in some markers (".0" <zero> vs. " . " <nan>). However, I cannot totally exclude a formatting issue when opening the .sav file. The file is saved in the .sav format of SPSS, which can be opened and converted by open source software as "PSPP", too. Providing a text file as .csv would however allow a direct import in R, Python, Excel etc and would prevent formatting errors (e.g. from guys like me).

Experimental design

No comment.

Validity of the findings

No comment.

Additional comments

The idea to dissect different lymphocytic phenotypes of the peritumoral area is much appreciated, which allows further correlations as well to define some kind of peritumoral immunological footprint (e.g. negative correlation between FoxP3 and ER etc.). This of course does not have to be considered but it could be and it would be already in your data.

---

## Round 0.2 · Minor Revisions

A few additional minor corrections were suggested by two reviewers.

Please make the relevant changes and re-upload.

Reviewer 2 ·

Basic reporting

The authors have addressed many points, however, they still neglect some of the latest findings - look at three studies in Nature 2020, by Cabrita et al., Petitprez et al. and Helmink et al..; or Sherene Loi KCO 2019.
sTILs at primary tumour site do influence outcome significantly - and there is a higher impact in patients with higher nodal disease.
The discussion needs to be structured, subheading would help to extract take-home messages.

Given the new addition " In this second multivariate model the immune populations of CD68+ macrophages were a factor associated with ALN metastasis (page 7, lines 181-188). " - should the authors now not revise their general statements and title:
“Peritumoral immune infiltrates in primary tumors are not associated with axillary lymph node metastasis in breast cancer: a retrospective cohort study"?

Experimental design

na

Validity of the findings

na

Additional comments

na

·

Basic reporting

Thank you for having addressed the issues mentioned. Please also correct the number of patients in the methods part of the abstract (still n = 150 vs. correct n = 149).

Experimental design

No comment.

Validity of the findings

No Comment.

---

## Round 0.3 · accepted · Accept

Please check carefully in the method. I believe the section on TMAs still says "150 patients." If this is incorrect and should be 149, please change in proofing.